# Minimal Random Code Learning: Getting Bits Back from Compressed Model Parameters

**Marton Havasi**
Department of Engineering
University of Cambridge
mh740@cam.ac.uk

**Robert Peharz**
Department of Engineering
University of Cambridge
rp587@cam.ac.uk

**José Miguel Hernández-Lobato**
Department of Engineering
University of Cambridge,
Microsoft Research,
Alan Turing Institute
jmh233@cam.ac.uk

## Abstract

While deep neural networks are a highly successful model class, their large memory footprint puts considerable strain on energy consumption, communication bandwidth, and storage requirements. Consequently, model size reduction has become an utmost goal in deep learning. A typical approach is to train a set of deterministic weights, while applying certain techniques such as pruning and quantization, in order that the empirical weight distribution becomes amenable to Shannon-style coding schemes. However, as shown in this paper, relaxing weight determinism and using a full variational distribution over weights allows for more efficient coding schemes and consequently higher compression rates. In particular, following the classical bits-back argument, we encode the network weights using a random sample, requiring only a number of bits corresponding to the Kullback-Leibler divergence between the sampled variational distribution and the encoding distribution. By imposing a constraint on the Kullback-Leibler divergence, we are able to explicitly control the compression rate, while optimizing the expected loss on the training set. The employed encoding scheme can be shown to be close to the optimal information-theoretical lower bound, with respect to the employed variational family. Our method sets new state-of-the-art in neural network compression, as it strictly dominates previous approaches in a Pareto sense: On the benchmarks LeNet-5/MNIST and VGG-16/CIFAR-10, our approach yields the best test performance for a fixed memory budget, and vice versa, it achieves the highest compression rates for a fixed test performance.

## 1 Introduction

With the celebrated success of deep learning models and their ever increasing presence, it has become a key challenge to increase their efficiency. In particular, the rather substantial memory requirements in neural networks can often conflict with storage and communication constraints, especially in mobile applications. Moreover, as discussed in Han et al. (2015), memory accesses are up to three orders of magnitude more costly than arithmetic operations in terms of energy consumption. Thus, compressing deep learning models has become a priority goal with a beneficial economic and ecological impact.

Traditional approaches to model compression usually rely on three main techniques: pruning, quantization and coding. For example, Deep Compression (Han et al., 2016) proposes a pipeline employing all three of these techniques in a systematic manner. From an information-theoretic perspective, the central routine is *coding*, while pruning and quantization can be seen as helper heuristics to reduce the entropy of the empirical weight-distribution, leading to shorter encoding lengths (Shannon, 1948). Also, the recently proposed Bayesian Compression (Louizos et al., 2017) falls into this scheme, despite being motivated by the so-called *bits-back* argument (Hinton & Van Camp, 1993) which theoretically allows for higher compression rates.[1] While the bits-back argument certainly

---

[1] Recall that the bits-back argument states that, assuming a large dataset and a neural network equipped with a weight-prior $p$, the effective coding cost of the network weights is $\mathrm{KL}(q||p) = \mathbb{E}_q[\log \frac{q}{p}]$, where $q$ is

motivated the use of variational inference in Bayesian Compression, the downstream encoding is still akin to Deep Compression (and other approaches). In particular, the variational distribution is merely used to derive a *deterministic set of weights*, which is subsequently encoded with Shannon-style coding. This approach, however, does not fully exploit the coding efficiency postulated by the bits-back argument.

In this paper, we step aside from the pruning-quantization pipeline and propose a novel coding method which approximately realizes bits-back efficiency. In particular, we refrain from constructing a deterministic weight-set but rather encode a *random weight-set* from the full variational posterior. This is fundamentally different from first drawing a weight-set and subsequently encoding it – this would be no more efficient than previous approaches. Rather, the coding scheme developed here is allowed to pick a random weight-set which can be cheaply encoded. By using results from Harsha et al. (2010), we show that such an coding scheme always exists and that the bits-back argument indeed represents a theoretical lower bound for its coding efficiency. Moreover, we propose a practical scheme which produces an approximate sample from the variational distribution and which can indeed be encoded with this efficiency. Since our algorithm learns a distribution over weight-sets and derives a random message from it, while minimizing the resulting code length, we dub it *Minimal Random Code Learning* (MIRACLE).

From a practical perspective, MIRACLE has the advantage that it offers *explicit control* over the expected loss and the compression size. This is distinct from previous techniques, which require tedious tuning of various hyper-parameters and/or thresholds in order to achieve a certain coding goal. In our method, we can simply control the KL-divergence using a penalty factor, which directly reflects the achieved code length (plus a small overhead), while simultaneously optimizing the expected training loss. As a result, we were able to trace the trade-off curve for compression size versus classification performance (Figure 1). We clearly outperform previous state-of-the-art in a Pareto sense: For any desired compression rate, our encoding achieves better performance on the test set; vice versa, for a certain performance on the test set, our method achieves the highest compression. To summarize, our main contributions are:

- We introduce MIRACLE, an innovative compression algorithm that exploits the noise resistance of deep learning models by training a variational distribution and efficiently encodes a random set of weights.
- Our method is easy to implement and offers explicit control over the loss and the compression size.
- We provide theoretical justification that our algorithm gets close to the theoretical lower bound on the encoding length.
- The potency of MIRACLE is demonstrated on two common compression tasks, where it clearly outperforms previous state-of-the-art methods for compressing neural networks.

In the following section, we discuss related work and introduce required background. In Section 3 we introduce our method. Section 4 presents our experimental results and Section 5 concludes the paper.

## 2 RELATED WORK

There is an ample amount of research on compressing neural networks, so that we will only discuss the most prominent ones, and those which are related to our work. An early approach is *Optimal Brain Damage* (LeCun et al., 1990) which employs the Hessian of the network weights in order to determine whether weights can be pruned without significantly impacting training performance. A related but simpler approach was proposed in Han et al. (2015), where small weights are truncated to zero, alternated with re-training. This simple approach yielded – somewhat surprisingly – networks which are one order of magnitude smaller, without impairing performance. The approach was refined into a systematic pipeline called *Deep Compression*, where magnitude-based weight pruning is followed by weight quantization (clustering weights) and Huffman coding (Huffman, 1952). While

---

a variational posterior. However, in order to realize this effective cost, one needs to encode *both* the network weights and the training targets, while it remains unclear whether it can also be achieved for network weights alone.

its compression ratio ($\sim 50\times$) has been surpassed since, many of the subsequent works took lessons from this paper.

*HashNet* proposed by Chen et al. (2015) also follows a simple and surprisingly effective approach: They exploit the fact that training of neural networks is resistant to imposing random constraints on the weights. In particular, they use hashing to enforce groups of weights to share the same value, yielding memory reductions of up to $64\times$ with gracefully degrading performance. *Weightless* encoding by Reagen et al. (2018) demonstrates that neural networks are resilient to weight noise, and exploits this fact for a lossy compression algorithm. The recently proposed *Bayesian Compression* (Louizos et al., 2017) uses a Bayesian variational framework and is motivated by the *bits-back argument* (Hinton & Van Camp, 1993). Since this work is the closest to ours, albeit with important differences, we discuss Bayesian Compression and the bits-back argument in more detail.

The basic approach is to equip the network weights $\boldsymbol{w}$ with a prior $p$ and to approximate the posterior using the standard variational framework, i.e. maximize the *evidence lower bound* (ELBO) for a given dataset $\mathcal{D}$

$$\mathbb{E}_{q_\phi}[\log p(\mathcal{D}|\boldsymbol{w})] - \mathrm{KL}(q_\phi||p)\,, \tag{1}$$

w.r.t. the variational distribution $q_\phi$, parameterized by $\phi$. The bits-back argument (Hinton & Van Camp, 1993) establishes a connection between the Bayesian variational framework and the *Minimum Description Length* (MDL) principle (Grünwald, 2007). Assuming a large dataset $\mathcal{D}$ of input-target pairs, we aim to use the neural network to transmit the *targets* with a minimal message, while the *inputs* are assumed to be public. To this end, we draw a weight-set $\boldsymbol{w}^*$ from $q_\phi$, which has been obtained by maximizing (1); note that knowing a particular weight $\boldsymbol{w}^*$ set conveys a message of length $\mathrm{H}[q_\phi]$ (H refers to the Shannon entropy of the distribution). The weight-set $\boldsymbol{w}^*$ is used to encode the residual of the targets, and is itself encoded with the prior distribution $p$, yielding a message of length $\mathbb{E}_{q_\phi}[-\log p(\mathcal{D}|\boldsymbol{w})] + \mathbb{E}_{q_\phi}[\log p]$. This message allows the receiver to perfectly reconstruct the original targets, and consequently the variational distribution $q_\phi$, by running the same (deterministic) algorithm as used by the sender. Consequently, with $q_\phi$ at hand, the receiver is able to retrieve an auxiliary message encoded in $\boldsymbol{w}^*$. When subtracting the length of this "free message" from the original $\mathbb{E}_{q_\phi}[\log p]$ nats,[2] we yield a net cost of $\mathrm{KL}(q_\phi||p) = \mathbb{E}_{q_\phi}[\log \frac{q_\phi}{p}]$ nats for encoding the weights, i.e. we recover the ELBO (1) as negative MDL (Hinton & Van Camp, 1993).

In (Hinton & Zemel, 1994; Frey & Hinton, 1997) coding schemes were proposed which practically exploited the bits-back argument for the purpose of coding *data*. However, it is not clear how these free bits can be spent solely for the purpose of model compression, as we only want to store a representation of our model, while discarding the training data. Therefore, while Bayesian Compression is certainly *motivated* by the bits-back argument, it actually does not strive for the postulated coding efficiency $\mathrm{KL}(q_\phi||p)$. Rather, this method imposes a sparsity inducing prior distribution to aid the pruning process. Moreover, high posterior variance is translated into reduced precision which constitutes a heuristic for quantization. In the end, Bayesian Compression merely produces a *deterministic* weight-set $\boldsymbol{w}^*$ which is encoded similar as in preceding works.

In particular, all previous approaches essentially use the following coding scheme, or a (sometimes sub-optimal) variant of it. After a deterministic weight-set $\boldsymbol{w}^*$ has been obtained, involving potential pruning and quantization techniques, one interprets $\boldsymbol{w}^*$ as a sequence of i.i.d. variables, taking values from a finite alphabet. Then one assumes the coding distribution $p'(w) = \frac{1}{N}\sum_{i=1}^N \delta_{w_i^*}(w)$, where $\delta_x$ denotes the Kronecker delta at $x$. According to Shannon's source coding theorem (Shannon, 1948), $\boldsymbol{w}^*$ can be coded with no less than $N\mathrm{H}[p']$ nats, which is asymptotically achieved by Huffman coding, like in Han et al. (2016). Note that the Shannon lower bound can be written as

$$N\mathrm{H}[p'] = -\sum_{i=1}^N \log p'(w_i^*) = -\log p'(\boldsymbol{w}^*) = \sum_{\boldsymbol{w}} \delta_{\boldsymbol{w}^*}(\boldsymbol{w}) \log \frac{\delta_{\boldsymbol{w}^*}(\boldsymbol{w})}{p'(\boldsymbol{w})} = \mathrm{KL}(\delta_{\boldsymbol{w}^*}||p'), \tag{2}$$

where we have set $p'(\boldsymbol{w}) = \prod_i p'(w_i)$. Thus, these Shannon-style coding schemes are in some sense optimal, when the variational family is *restricted to point-measures*, i.e. deterministic weights. By extending the variational family to comprise more general distributions $q$, the coding length $\mathrm{KL}(q||p)$ could be drastically reduced. In the following, we develop such a method which exploits the uncertainty represented by $q$ in order to encode a *random* weight-set with short coding length.

---

[2] Unless otherwise stated, we refer to information theoretic measures in nats. For reference, $1\,\mathrm{nat} = \log_2 e$ bits $\approx 1.44$ bits

## 3 MINIMAL RANDOM CODE LEARNING

Consider the scenario where we want to train a neural network but our memory budget is constrained to $C$ nats. As illustrated in the previous section, a variational approach offers – in principle – a simple and elegant solution. Before we proceed, we note that we do not consider our approach to be a strictly Bayesian one, but rather based on the MDL principle, although these two are of course highly related (Grünwald, 2007). In particular, we refer to $p$ as an *encoding distribution* rather than a prior, and moreover we will use a framework akin to the $\beta$-VAE (Higgins et al., 2017) which better reflects our goal of efficient coding. The crucial difference to the $\beta$-VAE being that we encode *parameters* rather than *data*.

Now, similar to Louizos et al. (2017), we first fix a suitable network architecture, select an encoding distribution $p$ and a parameterized variational family $q_\phi$ for the network weights $\boldsymbol{w}$. We consider, however, a slightly different variational objective related to the $\beta$-VAE:

$$\mathcal{L}(\phi) = \underbrace{\mathbb{E}_{q_\phi}[\log p(\mathcal{D}|\boldsymbol{w})]}_{\text{negative loss}} - \beta \underbrace{\text{KL}(q_\phi||p)}_{\text{model complexity}} . \tag{3}$$

This objective directly reflects our goal of achieving both a good training performance (loss term) and being able to represent our model with a short code (model complexity), at least according to the bits-back argument. After obtaining $q_\phi$ by maximizing (3), a weight-set drawn from $q_\phi$ will perform comparable to a deterministically trained network, since the variance of the negative loss term will be comparatively small to the mean, and since the KL term regularizes the model. Thus, our declared goal is to *draw a sample from $q_\phi$* such that this sample can be *encoded as efficiently as possible*. This problem can be formulated as the following communication problem.

Alice observes a training data set $(X, Y) = \mathcal{D}$ drawn from an unknown distribution $p(D)$. She trains a variational distribution $q_\phi(\boldsymbol{w})$ by optimizing (3) for a given $\beta$ using a deterministic algorithm. Subsequently, she wishes to send a message $M(\mathcal{D})$ to Bob, which allows him to generate a sample distributed according to $q_\phi$. How long does this message need to be?

The answer to this question depends on the unknown data distribution $p(D)$, so we need to make an assumption about it. Since the variational parameters $\phi$ depend on the realized dataset $\mathcal{D}$, we can interpret the variational distribution as a conditional distribution $q(\boldsymbol{w}|D) := q_\phi(\boldsymbol{w})$, giving rise to the joint $q(\boldsymbol{w}, D) = q(\boldsymbol{w}|D)p(D)$. Now, our assumption about $p(D)$ is that $\int q(\boldsymbol{w}|\mathcal{D})p(\mathcal{D})\,d\mathcal{D} = p(\boldsymbol{w})$, that is, the variational distribution $q_\phi$ yields the assumed encoding distribution $p(\boldsymbol{w})$, when averaged over all possible datasets. Note that this a similar strong assumption as in a Bayesian setting, where we assume that the data distribution is given as $p(D) = \int p(D|\boldsymbol{w})p(\boldsymbol{w})\,d\boldsymbol{w}$. In this setting, it follows immediately from the data processing inequality (Harsha et al., 2010) that in expectation the message length $|M|$ cannot be smaller than $\text{KL}(q_\phi||p)$:

$$\mathbb{E}_D[|M|] \geq \text{H}[M] \geq \text{I}[D : M] \geq \text{I}[D : \boldsymbol{w}] = \int \text{KL}(q(\boldsymbol{w}|\mathcal{D})||p(\boldsymbol{w}))\,d\mathcal{D} = \mathbb{E}_D[\text{KL}(q_\phi||p)], \tag{4}$$

where I refers to the mutual information and in the third inequality we applied the data processing inequality for Markov chain $D \to M \to \boldsymbol{w}$. As discussed by Harsha et al. (2010), the inequality $\mathbb{E}_D[|M|] \geq \mathbb{E}_D[\text{KL}(q_\phi||p)]$ can be very loose. However, as they further show, the message length can be brought close to the lower bound, *if Alice and Bob are allowed to share a source of randomness*:

**Theorem 3.1 (Harsha et al. (2010))** *Given random variables $D$, $\boldsymbol{w}$ and a random string $R$, let a protocol $\Pi$ be defined via a message function $M(D, R)$ and a decoder function $\boldsymbol{w}(M, R)$, i.e. $\Pi(D) = \boldsymbol{w}(M(D, R), R)$. Let $\text{T}_\Pi(D) := \mathbb{E}_R[|M(D, R)|]$ be the expected message length for data $D$, and let the minimal expected message length be defined as*

$$\text{T}[D : \boldsymbol{w}] := \min_\Pi \mathbb{E}_D[T_\Pi(D)], \tag{5}$$

*where $\Pi$ ranges over all protocols such that $D, \boldsymbol{w}$ and $D, \Pi(D)$ have the same distribution. Then*

$$\text{I}[D : \boldsymbol{w}] \leq \text{T}[D : \boldsymbol{w}] \leq \text{I}[D : \boldsymbol{w}] + 2\log(\text{I}[D : \boldsymbol{w}] + 1) + O(1). \tag{6}$$

The results of Harsha et al. (2010) establish a characterization of the mutual information in terms of minimal coding a conditional sample. For our purposes, Theorem 3.1 guarantees that in principle

---

**Algorithm 1** Minimal Random Coding

```
1: procedure ENCODE(q_φ, p)
2:     K ← exp(KL(q_φ||p))
3:     draw K samples {w_k}_{k=0}^{K-1}, w_k ~ p                    ▷ using shared random generator
4:     a_k ← q_φ(w_k)/p(w_k)
5:     q̃(w_k) := a_k / (∑_{k'=0}^{K-1} a_{k'})  for k ∈ {0...K-1}
6:     draw a sample w_{k*} ~ q̃
7:     return w_{k*}, k*
8: end procedure
```

---

there is an algorithm which realizes near bits-back efficiency. Furthermore, the theorem shows that this is indeed a fundamental lower bound, i.e. that such an algorithm is optimal for the considered setting. To this end, we need to refer to a "common ground", i.e. a shared random source $\mathcal{R}$, where w.l.o.g. we can assume that this source is an infinite list of samples from our encoding distribution $p$. In practice, this can be realized via a pseudo-random generator with a public seed.

### 3.1 THE BASIC ALGORITHM

While Harsha et al. (2010) provide a constructive proof using a variant of rejection sampling (see Appendix A), this algorithm is in fact intractable, because it requires keeping track of the acceptance probabilities over the whole sample domain. Therefore, we propose an alternative method to produce an approximate sample from $q_\phi$, depicted in Algorithm 1. This algorithm takes as inputs the trained variational distribution $q_\phi$ and the encoding distribution $p$. We first draw $K = \exp(\mathrm{KL}(q_\phi||p))$ samples from $p$, using the shared random generator. Subsequently, we craft a discrete proxy distribution $\tilde{q}$, which has support only on these $K$ samples, and where the probability mass for each sample is proportional to the importance weights $a_k = \frac{q_\phi(\boldsymbol{w}_k)}{p(\boldsymbol{w}_k)}$. Finally, we draw a sample from $\tilde{q}$ and return its index $k^*$ and the sample $\boldsymbol{w}_{k^*}$ itself. Since any number $0 \leq k^* < K$ can be easily encoded with $\mathrm{KL}(q_\phi||p)$ nats, we achieve our aimed coding efficiency. *Decoding* the sample is easy: simply draw the $k^{*\mathrm{th}}$ sample $\boldsymbol{w}_{k^*}$ from the shared random generator (e.g. by resetting the random seed).

While this algorithm is remarkably simple and easy to implement, there is of course the question of whether it is a correct thing to do. Moreover, an immediate caveat is that the number $K$ of required samples grows exponentially in $\mathrm{KL}(q_\phi||p)$, which is clearly infeasible for encoding a practical neural network. The first point is addressed in the next section, while the latter is discussed in Section 3.3, together with other practical considerations.

### 3.2 THEORETICAL ANALYSIS

The proxy distribution $\tilde{q}$ in Algorithm 1 is based on an importance sampling scheme, as its probability masses are defined to be proportional to the usual importance weights $a_k = \frac{q_\phi(\boldsymbol{w}_k)}{p(\boldsymbol{w}_k)}$. Under mild assumptions ($q_\phi$, $p$ continuous; $a_k < \infty$) it is easy to verify that $\tilde{q}$ converges to $q_\phi$ in distribution for $K \to \infty$; thus in the limit, Algorithm 1 samples from the correct distribution. However, since we collect only $K = \exp(\mathrm{KL}(q_\phi||p))$ samples in order to achieve a short coding length, $\tilde{q}$ will be biased. Fortunately, it turns out that $K$ is just in the right order for this bias to be small.

**Theorem 3.2 (Low Bias of Proxy Distribution)** *Let $q_\phi$, $p$ be distributions over $\boldsymbol{w}$. Let $t \geq 0$ and $\tilde{q}$ be a discrete distribution constructed by drawing $K = \exp(\mathrm{KL}(q_\phi||p) + t)$ samples $\{\boldsymbol{w}_k\}_{k=0}^{K-1}$ from $p$ and defining $\tilde{q}(\boldsymbol{w}_k) := \frac{q_\phi(\boldsymbol{w}_k)/p(\boldsymbol{w}_k)}{\sum_{k'} q_\phi(\boldsymbol{w}_{k'})/p(\boldsymbol{w}_{k'})}$. Furthermore, let $f(\boldsymbol{w})$ be a measurable function and $||f||_{q_\phi} = \sqrt{\mathbb{E}_{q_\phi}[f^2]}$ be its 2-norm under $q_\phi$. Then it holds that*

$$\mathbb{P}\left(\left|\mathbb{E}_{\tilde{q}}[f] - \mathbb{E}_{q_\phi}[f]\right| \geq \frac{2||f||_{q_\phi}\epsilon}{1-\epsilon}\right) \leq 2\epsilon \tag{7}$$

*where*

$$\epsilon = \left(e^{-t/4} + 2\sqrt{\mathbb{P}\left(\log(q_\phi/p) > \mathrm{KL}(q_\phi||p) + t/2\right)}\right)^{1/2}. \tag{8}$$

---

**Algorithm 2** Minimal Random Code Learning (MIRACLE)

---

1: **procedure** LEARN($\mathcal{D}$, model with parameters $\boldsymbol{w}$, $C$, $C_{loc}$, $I_0$, $I$)
2:     randomly split $\boldsymbol{w}$ into $B = \lceil \frac{C}{C_{loc}} \rceil$ blocks $\{\boldsymbol{w}_0, \ldots \boldsymbol{w}_{B-1}\}$
3:     $\mathcal{O} \leftarrow \{0, \ldots, B-1\}$                          ▷ The blocks that have not yet been encoded
4:     $\beta_b \leftarrow \epsilon_{\beta 0}$, for $b \in \{0, \ldots, B-1\}$
5:     VARIATIONAL UPDATES($I_0$)
6:     **while** $\mathcal{O} \neq \emptyset$ **do**
7:         draw random $b$ from $\mathcal{O}$
8:         $\mathcal{O} \leftarrow \mathcal{O} \setminus \{b\}$
9:         $\boldsymbol{w}_b^*, k_b = \text{ENCODE}(q_\phi(\boldsymbol{w}_b), p(\boldsymbol{w}_b))$               ▷ from Algorithm 1
10:        $\boldsymbol{w}_b \leftarrow \boldsymbol{w}_b^*$ (fixing the value of $\boldsymbol{w}_b$)
11:        VARIATIONAL UPDATES($I$)
12:     **end while**
13:     **return** $[k_0, \ldots, k_{B-1}]$
14: **end procedure**

15: **procedure** VARIATIONAL UPDATES($I$)
16:     $\mathcal{L}_\mathcal{O} := \mathbb{E}_{q_\phi(\{\boldsymbol{w}_b\}_{b \in \mathcal{O}})}[\log p(\mathcal{D}|\boldsymbol{w})] - \sum_{b \in \mathcal{O}} \beta_b \text{KL}(q_\phi(\boldsymbol{w}_b)||p(\boldsymbol{w}_b))$
17:     **for** $i \in [0, \ldots, I-1]$ **do**
18:         Perform stochastic gradient update of $\mathcal{L}_\mathcal{O}$
19:         **for** $b \in \mathcal{O}$ **do**
20:             **if** $\text{KL}(q_\phi(\boldsymbol{w}_b)||p(\boldsymbol{w}_b)) > C_{loc}$ **then**
21:                 $\beta_b \leftarrow (1 + \epsilon_\beta) \times \beta_b$
22:             **else**
23:                 $\beta_b \leftarrow \beta_b / (1 + \epsilon_\beta)$
24:             **end if**
25:         **end for**
26:     **end for**
27: **end procedure**

---

Theorem 3.2 is a corollary of Chatterjee & Diaconis (2018), Theorem 1.2, by noting that

$$\mathbb{E}_{\tilde{q}}[f] = \frac{1}{\sum_{k'} \frac{q_\phi(\boldsymbol{w}_{k'})}{p(\boldsymbol{w}_{k'})}} \sum_k f(\boldsymbol{w}_k) \frac{q_\phi(\boldsymbol{w}_k)}{p(\boldsymbol{w}_k)}, \tag{9}$$

which is precisely the importance sampling estimator for unnormalized distributions (denoted as $J_n$ in (Chatterjee & Diaconis, 2018)), i.e. their Theorem 1.2 directly yields Theorem 3.2. Note that the term $e^{-t/4}$ decays quickly with $t$, and, since $\log q_\phi/p$ is typically concentrated around its expected value $\text{KL}(q||p)$, the second term in (8) also quickly becomes negligible. Thus, roughly speaking, Theorem 3.2 establishes that $\mathbb{E}_{q_\phi}[f] \approx \mathbb{E}_{\tilde{q}}[f]$ with high probability, for any measurable function $f$. This is in particular true for the function $f(\boldsymbol{w}) = \log p(\mathcal{D}|\boldsymbol{w}) - \beta \log \frac{q_\phi(\boldsymbol{w})}{p(\boldsymbol{w})}$. Note that the expectation of this function is just the variational objective (3) we optimized to yield $q_\phi$ in the first place. Thus, since $\mathbb{E}_{\tilde{q}}[f] \approx \mathbb{E}_{q_\phi}[f] = \mathcal{L}(\phi)$, replacing $q_\phi$ by $\tilde{q}$ is well justified. Thereby, any sample of $\tilde{q}$ can trivially be encoded with $\text{KL}(q_\phi||p)$ nats, and decoded by simple reference to a pseudo-random generator.

Note that according to Theorem 3.2 we should actually take a number of samples somewhat larger than $\exp(\text{KL}(q_\phi||p))$ in order to make $\epsilon$ sufficiently small. In particular, the results in (Chatterjee & Diaconis, 2018) also imply that a too small number of samples will typically be quite off the targeted expectation (for the worst-case $f$). However, although our choice of number of samples is at a critical point, in our experiments this number of samples yielded very good results.

### 3.3 PRACTICAL IMPLEMENTATION

In this section, we describe the application of Algorithm 1 within a practical learning algorithm – Minimal Random Code Learning (MIRACLE) – depicted in Algorithm 2. For both $q_\phi$ and $p$ we used Gaussians with diagonal covariance matrices. For $q_\phi$, all means and standard deviations constituted

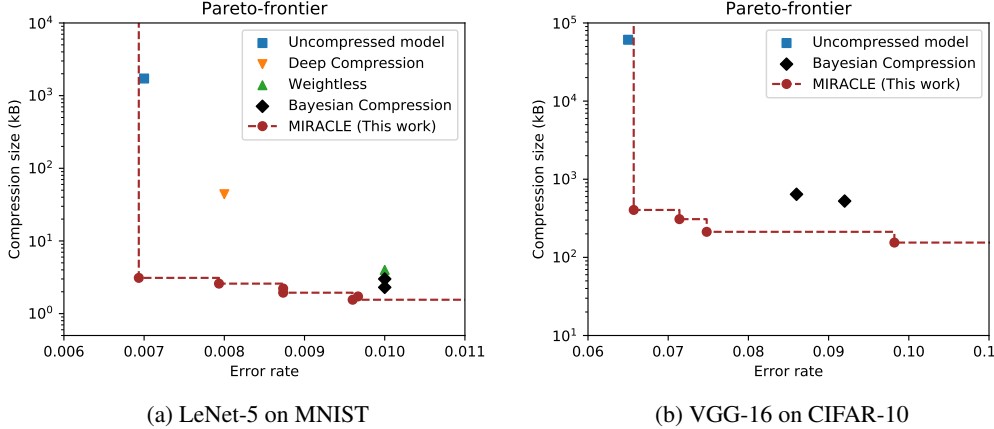

Figure 1: The error rate and the compression size for various compression methods. Lower left is better.

the variational parameters $\phi$. The mean of $p$ was fixed to zero, and the standard deviation was shared within each layer of the encoded network. These shared parameters of $p$ where learned jointly with $q_\phi$, i.e. the encoding distribution was also adapted to the task. This choice of distributions allowed us to use the reparameterization trick for effective variational training and furthermore, $\mathrm{KL}(q_\phi||p)$ can be computed analytically.

Since generating $K = \exp(\mathrm{KL}(q_\phi||p))$ samples is infeasible for any reasonable $\mathrm{KL}(q_\phi||p)$, we divided the overall problem into sub-problems. To this end, we set a *global coding goal* of $C$ nats and a *local coding goal* of $C_{loc}$ nats. We randomly split the weight vector $\boldsymbol{w}$ into $B = \lceil \frac{C}{C_{loc}} \rceil$ equally sized blocks, and assigned each block an allowance of $C_{loc}$ nats. For example, fixing $C_{loc}$ to $11.09$ nats $\approx 16$ bits, corresponds to $K = 65536$ samples which need to be drawn per block. We imposed *block-wise* KL constraints using block-wise penalty factors $\beta_b$, which were automatically annealed via multiplication/division with $(1 + \epsilon_\beta)$ during the variational updates (see Algorithm 2). Note that the random splitting into $B$ blocks can be efficiently coded via the shared random generator, and only the number $B$ needs communicated.

Before encoding any weights, we made sure that variational learning had converged by training for a large number of iterations $I_0 = 10^4$. After that, we alternated between encoding single blocks and updating the variational distribution not-yet coded weights, by spending $I$ intermediate variational iterations. To this end, we define a variational objective $\mathcal{L}_\mathcal{O}$ w.r.t. to blocks which have not been coded yet, while weights of already encoded blocks were fixed to their encoded value. Intuitively, this allows to compensate for poor choices in earlier encoded blocks, and was crucial for good performance. Theoretically, this amounts to a rich auto-regressive variational family $q_\phi$, as the blocks which remain to be updated are effectively conditioned on the weights which have already been encoded. We also found that the hashing trick (Chen et al., 2015) further improves performance (not depicted in Algorithm 2 for simplicity). The hashing trick randomly conditions weights to share the same value. While Chen et al. (2015) apply it to reduce the entropy, in our case it helps to restrict the optimization space and reduces the dimensionality of both $p$ and $q_\phi$. We found that this typically improves the compression rate by a factor of $\sim 1.5\times$.

## 4 EXPERIMENTAL RESULTS

The experiments[3] were conducted on two common benchmarks: LeNet-5 on MNIST and VGG-16 on CIFAR-10. As baselines we used three recent state-of-the-art methods, namely Deep Compression (Han et al., 2016), Weightless encoding (Reagen et al., 2018) and Bayesian Compression (Louizos et al., 2017). The performance of the baseline methods are quoted from their respective source materials.

---

[3] The code is publicly available at `https://github.com/cambridge-mlg/miracle`

Table 1: Numerical performance of the compression algorithms.

| Model | Compression | Size | Ratio | Test error |
|---|---|---|---|---|
| LeNet-5 on MNIST | Uncompressed model | 1720 kB | 1× | 0.7 % |
| | Deep Compression | 44 kB | 39× | 0.8 % |
| | Weightless [5] | 4.52 kB | 382× | 1.0 % |
| | Bayesian Compression | 2.3 kB | 771× | 1.0 % |
| | MIRACLE (Lowest error) | 3.03 kB | 555× | **0.69 %** |
| | MIRACLE (Highest compression) | 1.52 kB | **1110×** | 0.96 % |
| VGG-16 on CIFAR-10 | Uncompressed model | 60 MB | 1× | 6.5 % |
| | Bayesian Compression | 642 kB | 95× | 8.6 % |
| | Bayesian Compression | 525 kB | 116× | 9.2 % |
| | MIRACLE (Lowest error) | 417 kB | 147× | **6.57 %** |
| | MIRACLE (Highest compression) | 168 kB | **365×** | 10.0 % |

For training MIRACLE, we used Adam (Kingma & Ba, 2014) with the default learning rate ($10^{-3}$) and we set $\epsilon_{\beta 0} = 10^{-8}$ and $\epsilon_\beta = 5 \times 10^{-5}$. For VGG, the means of the weights were initialized using a pretrained model.[4] We recommend applying the hashing trick mainly to reduce the size of the largest layers. In particular, we applied the hashing trick was to layers 2 and 3 in LeNet-5 to reduce their sizes by $2\times$ and $64\times$ respectively and to layers 10-16 in VGG to reduce their sizes $8\times$. The local coding goal $C_{loc}$ was fixed at 20 bits for LeNet-5 and it was varied between 15 and 5 bits for VGG ($B$ was kept constant). For the number of intermediate variational updates $I$, we used $I = 50$ for LeNet-5 and $I = 1$ for VGG, in order to keep training time reasonable ($\approx 1$ day on a single NVIDIA P100 for VGG).

The performance trade-offs (test error rate and compression size) of MIRACLE along with the baseline methods and the uncompressed model are shown in Figure 1 and Table 1. For MIRACLE we can easily construct the Pareto frontier, by starting with a large coding goal $C$ (i.e. allowing a large coding length) and successively reducing it. Constructing such a Pareto frontier for other methods is delicate, as it requires re-tuning hyper-parameters which are often only indirectly related to the compression size – for MIRACLE it is directly reflected via the KL-term. We see that MIRACLE is Pareto-better than the competitors: for a given test error rate, we achieve better compression, while for a given model size we achieve lower test error.

## 5    CONCLUSION

In this paper we followed through the philosophy of the bits-back argument for the goal of coding model parameters. The basic insight here is that restricting to a single deterministic weight-set and aiming to coding it in a classic Shannon-style is greedy and in fact sub-optimal. Neural networks – and other deep learning models – are highly overparameterized, and consequently there are many "good" parameterizations. Thus, rather than focusing on a single weight set, we showed that this fact can be exploited for coding, by selecting a "cheap" weight set out of the set of "good" ones. Our algorithm is backed by solid recent information-theoretic insights, yet it is simple to implement. We demonstrated that the presented coding algorithm clearly outperforms previous state-of-the-art. An important question remaining for future work is how efficient MIRACLE can be made in terms of memory accesses and consequently for energy consumption and inference time. There lies clear potential in this direction, as any single weight can be recovered by its block-index and relative index within each block. By smartly keeping track of these addresses, and using pseudo-random generators as algorithmic lookup-tables, we could design an inference machine which is able to directly run our compressed models, which might lead to considerable savings in memory accesses.

---

[4] For preprocessing the data and pretraining, we followed an open source implementation that can be found at `https://github.com/chengyangfu/pytorch-vgg-cifar10`

[5] Weighless encoding only reports the size of the two largest layers so we assumed that the size of the rest of the network is negligible in this case.

## ACKNOWLEDGEMENTS

We want to thank Christian Steinruecken, Olivér Janzer, Kris Stensbo-Smidt and Siddharth Swaroop for their helpful comments. This project has received funding from the European Union's Horizon 2020 research and innovation programme under the Marie Skłodowska-Curie Grant Agreement No. 797223 — HYBSPN. Furthermore, we acknowledge EPSRC and Intel for their support.

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

---

**Algorithm 3** Greedy Rejection Sampling

1: **procedure** SAMPLE($q$, $p$)
2:     $p_0(\boldsymbol{w}) \leftarrow 0$ for $\boldsymbol{w} \in \mathcal{W}$
3:     $p_0^* \leftarrow 0$
4:     **for** $i \leftarrow 0$ to $\infty$ **do**
5:         $\alpha_i(\boldsymbol{w}) \leftarrow \min\{q(\boldsymbol{w}) - p_{i-1}(\boldsymbol{w}), (1 - p_{i-1}^*)p(\boldsymbol{w})\}$
6:         $p_i(\boldsymbol{w}) \leftarrow p_{i-1}(\boldsymbol{w}) + \alpha_i(\boldsymbol{w})$
7:         $p_i^* \leftarrow \sum_{\boldsymbol{w} \in \mathcal{W}} p_i(\boldsymbol{w})$
8:         draw sample $\boldsymbol{w}_i \sim p$
9:         $\beta_i \leftarrow \frac{\alpha_i(\boldsymbol{w}_i)}{(1-p_{i-1}^*)p(\boldsymbol{w}_i)}$
10:         draw $\epsilon \sim \mathcal{U}(0, 1)$
11:         **if** $\epsilon \leq \beta_i$ **then**
12:             **return** $\boldsymbol{w}_i, i$
13:         **end if**
14:     **end for**
15: **end procedure**

---

## A GREEDY REJECTION SAMPLING BY HARSHA ET AL. (2010)

In order to prove the upper bound, to which Harsha et al. (2010) refer as the 'one-shot reverse Shannon theorem', they exhibit a rejection sampling procedure. However, instead of using the classical rejection with acceptance probabilities $\frac{q}{Mp}$ where $M = \max \frac{q}{p}$, they propose a greedier version. The core idea is that every sample should be accepted with as high probability as possible while keeping the overall acceptance probability of each element below the target distribution.

For this algorithm we assume discrete $p$ and $q$ over the set $\mathcal{W}$ and an infinite sequence of samples $\{\boldsymbol{w}_i\}_{i=1}^{\infty}$ from $p$.

Let $\alpha_i(\boldsymbol{w})$ with $i \in \mathrm{N}$ and $\boldsymbol{w} \in \mathcal{W}$ be the probability that the procedure outputs the $i$th sample with $\boldsymbol{w}_i = \boldsymbol{w}$. For the sampling method to be unbiased, we have to ensure that

$$q(\boldsymbol{w}) = \sum_{i=0}^{\infty} \alpha_i(\boldsymbol{w}) \,. \tag{10}$$

Let $p_i(\boldsymbol{w}) = \sum_{j=0}^{i} \alpha_j(\boldsymbol{w})$ be the probability that the procedure halts within $j \leq i$ iteration and it outputs $\boldsymbol{w}_j = \boldsymbol{w}$. Let $p_i^* = \sum_{\boldsymbol{w} \in \mathcal{W}} p_i(\boldsymbol{w})$ be the probability that procedure halts within $i$ iterations. Let

$$\begin{aligned} \alpha_i(\boldsymbol{w}) &= \min\{q(\boldsymbol{w}) - p_{i-1}(\boldsymbol{w}), (1 - p_{i-1}^*)p(\boldsymbol{w})\} \\ p_i(\boldsymbol{w}) &= p_{i-1}(\boldsymbol{w}) + \alpha_i(\boldsymbol{w}) \,. \end{aligned} \tag{11}$$

Since $P(\boldsymbol{w}_i = \boldsymbol{w}) = p(\boldsymbol{w})$, $\alpha_i(\boldsymbol{w})$ can be at most $(1 - p_{i-1}^*)p(\boldsymbol{w})$. The proposed strategy is greedy because it accepts the $i$th sample with as high probability as possible under the constraint that $p_i(\boldsymbol{w}) \leq q(\boldsymbol{w})$.

Under the proposed formula for $\alpha_i(\boldsymbol{w})$, the acceptance probability for the $i$th sample $\boldsymbol{w}_i$ is

$$\beta_i = \frac{\alpha_i(\boldsymbol{w}_i)}{(1 - p_{i-1}^*)p(\boldsymbol{w})} \tag{12}$$

The pseudo code is shown in Algorithm 3. Note that the algorithm requires computing $\alpha_i(\boldsymbol{w})$ for the whole set $\mathcal{W}$ in every iteration which makes it intractable for large $\mathcal{W}$.

### A.1 PROOF OUTLINE

For the details of the proof, please refer to the source material (Harsha et al., 2010).

To show that the procedure is unbiased, one has to prove that

$$q(\boldsymbol{w}) = \lim_{i \to \infty} p_i(\boldsymbol{w}) \,. \tag{13}$$

This is shown by proving that $q(\boldsymbol{w}) - p_i(\boldsymbol{w}) \leq q(\boldsymbol{w})(1 - p(\boldsymbol{w}))^i$ for $i \in \mathrm{N}$.

In order to bound the encoding length, one has to first show that if the accepted sample has index $i*$, then

$$\mathrm{E}[\log i^*] \leq \mathrm{KL}(q||p) + O(1) \,. \tag{14}$$

Following this, one can employ the prefix-free binary encoding of Vitanyi & Li (1997). Let $l(n)$ be the length of the encoding for $n \in \mathrm{N}$ using the encoding scheme proposed by Vitanyi & Li (1997). Their method is proven to have $|l(n)| = \log n + 2 \log \log(n + 1) + O(1)$, from which the upper bound follows:

$$\mathrm{T}^{\mathcal{R}}[D : W] \leq \mathrm{E}|l(i^*)| \leq \mathrm{KL}(q||p) + 2 \log(\mathrm{KL}(q||p) + 1) + O(1) \,. \tag{15}$$

