# OpenReview forum: "Minimal Random Code Learning: Getting Bits Back from Compressed Model Parameters"
_ICLR.cc/2019/Conference_

### Official Review · AnonReviewer3 · 2018-11-03
**The proposed approach has interesting formulation and good performance tradeoff while the main theorems are based on existing works.**

**Rating:** 7
**Confidence:** 3

**Review:**

This paper considers the compression of the model parameters in deep neural networks. The authors propose minimal random code learning (MIRACLE), which uses a random sample of weights and the variational framework interpreted by the bits-back argument. The authors introduce two theorems characterizing the properties of MIRACLE, and demonstrate its compression performance through the experiments.

The proposed approach is interesting and the performance on the benchmarks is good enough to demonstrate its effectiveness. However, since the two main theorems are based on the existing results by Harsha et al. (2010) and Chatterjee & Diaconis (2018), the main technical contribution of this paper is the sampling scheme in Algorithm 1.

Although the authors compare the performance trade-offs of MIRACLE with that of the baseline methods quoted from source materials, isn't it possible or desirable to include other competitors or other results for the baseline methods? Are there any other methods, in particular, achieving low error rate (with high compression size)? Little is discussed on why the baseline results are only a few.

minor comment:
- Eq.(4) lacks p(D) in front of dD.

Pros:
- Interesting approach based-on the bits back argument
- Good performance trade off demonstrated through experiments
Cons:
- Only a few baseline results, in particular, at high compression size

---

> ### Author Response · Authors · 2018-11-19
> **Reply**
>
> Thank you for your review and the constructive feedback.
>
> A point of criticism raised in the review was that we built on existing works. However, we want to point out that Harsha et al. (2010) presented their work in a very different setting, i.e. a general bound on communication complexity. They provide a mathematical proof for the bound, which, however, does not translate to a feasible algorithm. We, on the other hand, propose a novel algorithm that achieves the bound on the encoding length and we provide performance guarantees using results from Chatterjee & Diaconis (2018) -- their results also do not directly translate to our setting (we consider an approximate sampling algorithm while they discuss the sample size required for importance sampling). As far as we know, neither of these works have received considerable attention in the machine learning literature.
>
> In terms of baselines, our main goal was to compare against Bayesian compression, a state-of-the-art algorithm that is also motivated by the bits-back argument. We included further state-of-the-art baselines where they were available (VGG16/CIFAR10 was only reported in the Bayesian compression paper). We omitted compression algorithms that focus on improving the efficiency at runtime since these typically have significantly worse performance in terms of compression. However, exploiting our compression scheme for efficient inference (time, energy) is an important direction we are currently following.

---

### Official Review · AnonReviewer1 · 2018-11-03
**Interesting argument**

**Rating:** 6
**Confidence:** 2

**Review:**

In this paper the authors propose to use MLD principle to encode the weights of NNs and still preserve the performance of the original network. The main comparison is from Han 2016, in which the authors use ad-hoc techniques to zero some coefficient and prune some connection + Huffman coding. In this case , the authors uses as a regularizer (See equation 3) a constraints that the weights are easy to compress. The results seem significant improvement with respect to the state of the art.

---

> ### Author Response · Authors · 2018-11-19
> **Summary of the contributions**
>
> Thank you for your review and the constructive feedback.
>
> Indeed, the main contributions of our paper are:
> 1) we propose a novel coding scheme for compressing neural network weights; in particular, we improve over the previously ubiquitous pruning-quantization pipeline and Shannon-style coding.
> 2) our algorithm achieves the theoretical lower bound predicted by theory (based on Harsha et al.) and achieves the efficiency predicted by Hinton et al.'s bits-back argument.
> 3) our algorithm allows, in contrast to previous work, to explicit control the trade-off between prediction quality and compression rate; in particular, please see our derived trade-off curves on Figure 1.

---

### Official Review · AnonReviewer2 · 2018-11-06
**Very interesting paper**

**Rating:** 7
**Confidence:** 4

**Review:**

The authors come up with a surprisingly elegant algorithm ("minimal random coding") which encodes samples from a posterior distribution, only using a number of bits that approximates the KL divergence between posterior and prior, while Shannon-type algorithms can only do this if the posterior is deterministic (a delta distribution). It can also be directly used to sample from continuous distributions, while Shannon-type algorithms require quantization. In my opinion, this is the main contribution of the paper.

The other part of the paper that is specifically concerned with weight compression ("MIRACLE") turns out to be a lot less elegant. It is somewhat ironic that the authors specifically call attention to the their algorithm sending random samples, as opposed to existing algorithms, which quantize and then send deterministic variables. This is clearly true for the basic algorithm, but, if I understand correctly, not for MIRACLE. It seems clear that neural networks are sensitive to random resampling of their weights -- otherwise, the authors would not have to fix the weights in each block and then do further gradient descent for the following blocks. What would happen if the distributions were held constant, and the algorithm would be run again, just with a different (but identical) random seed in both sender and receiver? It seems this would lead to a performance drop, demonstrating that (albeit indirectly), MIRACLE also makes a deterministic choice of weights.

Overall, I find the paper somewhat lacking in terms of evaluation. MIRACLE consists of a lot of parts. It is hard to assess how much of the final coding gain presented in table 1 is due to the basic algorithm. What is the effect of selecting other probability models, possibly different ones than Gaussians? Choosing appropriate distributions can have a significant impact on the value of the KL divergence. Exactly how much is gained by applying the hashing trick? Are the standard deviations of the priors included in the size, and how are they encoded?

This could be assessed more clearly by directly evaluating the basic algorithm. Theorem 3.2 predicts that the approximation error of algorithm 1 asymptotically zero, i.e. one can gain an arbitrarily good approximation to the posterior by spending more bits. But how many more are practically necessary? It would be fantastic to actually see some empirical data quantifying how large the error is for different distributions (even simple toy distributions). What are the worst vs. best cases?

---

> ### Author Response · Authors · 2018-11-19
> **Reply**
>
> Thank you for the constructive feedback. We hope that this reply addresses some of the concerns.
>
> Indeed, the paper has two key contributions. Firstly, the theoretical results that show that MIRACLE is a generalization of the Shannon-type coding schemes and secondly, the implementation of the algorithm that achieve state-of-the-art results on the two benchmark tasks.
>
> Reviewer 2 points out that, while the basic algorithm encodes random samples, this is not the case for MIRACLE since it fixes the weights in each block and does further training on the rest. We argue that they are still random samples because by retraining after fixing each block, we are no longer sampling from a mean-field Gaussian, we are effectively sampling from a more flexible, autoregressive distribution (since each dimension is dependent on the previous ones). Indeed, if the retraining step was omitted, then we would be sampling from a less flexible, mean-field Gaussian distribution which leads to worse performance.
>
> Regarding the point that the origin of the gain is unclear, we can provide rough estimates. The difference in compression size between a mean-field Gaussian (without retraining) and the flexible distribution with retraining is about 2 times. The use of the hashing trick gives an improvement of about 1.5 times.
>
> The size of the encoding of the standard deviation of the prior is trivial compared to the overall compression size. They take 32x(number of layers) bits in the final message. For LeNet-5, this is approximately 1.0 % of the overall compression size and for VGG-16, it is approximately 0.05 % of the overall compression size.
>
> Using different distributions is an interesting avenue to explore. We believe, for example, that sparsity inducing priors could be beneficial. In this version of the paper, we settled with Gaussians because the reparameterization trick straight forwardly applies and the KL divergence has a closed form.
>
> Regarding the approximation error of algorithm 1, we have not done extensive experiments to quantify it. In our experience, exp(KL) samples perform well enough. Following the suggestion in the review, we ran an experiment on a toy example to see how the approximation error changes with the number of samples. It show that the total variation tends to 0 as the number of samples K increases, but it is still difficult to quantify how K affects the overall performance. Toy experiment link: https://imgur.com/a/oadzAT2

---

> > ### Comment · AnonReviewer2 · 2018-11-29
> > **No score change**
> >
> > Thank you for this additional information. I believe the paper would benefit from including some of it in the camera-ready version.
> >
> > Based on the discussion, I'm keeping my original score.

---

### Public Comment · ~James_Townsend1 · 2018-10-12
**Cool paper!**

I enjoyed reading this a lot, thanks for producing it.

I have a question/suggestion. It seems that your method makes quantization unnecessary. It's known that in the standard bits back method, weights can be communicated at arbitrarily high precision without affecting the communication cost (see e.g. [1] p. 353). The theory in [2] which you apply also seems to hold for continuous random variables. Moreover you don't do seem to do quantization in your experiments (I may be missing something since I don't know TensorFlow well), so I assume you are already aware of this fact.

I'm wondering if it might be worth mentioning in your paper, perhaps in the related work section, that quantization is effectively optional with your method. This saves the person using your algorithm coding effort, and saves them from having to choose a heuristic quantization method.

[1] David Mackay, 2003: Information Theory, Inference and Learning Algorithms
[2] Sourav Chatterjee and Persi Diaconis, 2017: The Sample Size Required in Importance Sampling

---

> ### Author Response · Authors · 2018-10-12
> **Continuous distributions are easier to train**
>
> Thank you for your comment, we are glad that you found our work interesting.
>
> Indeed it is true that the method does not require quantization. Perhaps we were not too explicit about this, but all of our results hold for continuous distributions. In particular, we used continuous (Gaussian) distributions in our experiments, which made optimization of the variational objective easy.
>
> In earlier concepts, we actually did attempt to use quantization, resulting in discrete distributions. However, these models proved too difficult to optimize (we tried various gradient estimators such as Gumbel-Softmax, reinforce, straight through estimator), and we were unable to reach state-of-the-art performance with these approaches. This problem did not pertain to continuous distributions. A continuous distribution is straight-forward to train using SGD and the reparameterization trick. The training is stable and gives good performance.

---

### Public Comment · ~James_Townsend1 · 2018-11-14
**Question**

In step 6 of Algorithm 1 in your paper, the sender draws a sample from the distribution \tilde{q}. Could we make an argument, analogous to the standard bits back argument, that bits used to generate that sample could, in principle, be recovered by the receiver?

The number of bits used is the entropy of \tilde{q}. I'm not sure how to estimate that quantity or if/when it would be large enough to be significant.

---

> ### Author Response · Authors · 2018-11-19
> **Thank you for the question.**
>
> Yes, in principle one could recover the randomness used to sample from \tilde{q} analogous to the bits-back argument (although it would require sharing the training set inputs and errors). However, in our experience, most of the probability mass is concentrated in one sample meaning that the entropy is close to 0 so the gains are unlikely to be significant. It would be interesting to see how the entropy of \tilde{q} changes as the number of samples K increases. We expect that the entropy would be close to (log K - KL(q||p)).
>
> Edited.

---

### Meta-Review · Area_Chair1 · 2018-12-13
**Good paper**

**Confidence:** 4
**Recommendation:** Accept (Poster)

**Metareview:**

This paper proposes a novel coding scheme for compressing neural network weights using Shannon-style coding and a variational distribution over weights.  This approach is shown to improve over existing schemes for LeNet-5 on MNIST and VGG-16 on CIFAR-10, strictly dominating them in terms of compression/error rate tradeoffs. Comparing to more baselines would have been helpful. Theoretical analysis based on non-trivial extensions of prior work by Harsha et al. (2010) and  Chatterjee & Diaconis (2018) is also presented. Overall, there was consensus among the reviewers that the paper makes a solid contribution and should be published.